# Optimizing facility location, sizing, and growth time for a cultivated resource: A case study in coral aquaculture

**Ryu B. Lippmann**[1]*, **Kate J. Helmstedt**[1,2], **Mark T. Gibbs**[3], **Paul Corry**[1,2]

**1** School of Mathematical Sciences, Queensland University of Technology, Brisbane, Queensland, Australia,
**2** Centre for Data Science, Queensland University of Technology, Brisbane, Queensland, Australia,
**3** Australian Institute of Marine Science, Brisbane, Queensland, Australia

\* r.lippmann@qut.edu.au

**Data Availability Statement:** All relevant data are within the manuscript and its Supporting Information files.

## Abstract

Production of cultivated resources require additional planning that takes growth time into account. We formulate a mathematical programming model to determine the optimal location and sizing of growth facilities, impacted by resource survival rate as a function of its growth time. Our method informs strategic decisions regarding the number, location, and sizing of facilities, as well as operational decisions of optimal growth time for a cultivated resource in a facility to minimize total costs. We solve this facility location and sizing problem in the context of coral aquaculture for large-scale reef restoration using a two-stage algorithm and a linear mixed-integer solver. We assess growth time in a facility in terms of its impact on survival (post-deployment) considering growth quantity requirements and growth facility production constraints. We explore the sensitivity of optimal facility number, location, and sizing to changes in the geographic distribution of demand and cost parameters computationally. Results show that the relationship between growth time and survival is critical to optimizing operational decisions for grown resources. These results inform the value of data certainty to optimize the logistics of coral aquaculture production.

## 1 Introduction

Planning cultivated resource production presents added complexities to that of manufactured goods. Cultivated resources are grown and fostered for some amount of time before they are harvested or deployed. Decisions regarding facility location, facility sizing, and growth time ultimately determine the success of large-scale projects in producing and deploying grown resources [1]. This is particularly challenging for resources with two growth phases: first inside a facility, and then continued growth after release (e.g., trees grown for reforestation [2–4], coral grown for reef restoration [5, 6], or animals reared for wild release [7]). In designing such facilities, a pertinent question is how long to keep resources within the facility, which significantly affects production optimality. A trade-off arises: keeping resources in the facility for longer may increase post-release success but increases in-process inventory and associated operating costs. Whilst this oversimplification of growth time being correlated with

**Funding:** R.B.L. is supported by the Reef Restoration and Adaptation Program Logistics Scholarship. K.J.H. is supported by Australian Research Council Fellowship under Grant DE200101791. M.T.G. and P.C. were funded through the Reef Restoration and Adaptation Program, which is funded by the Reef Trust, a partnership between the Australian Government and the Great Barrier Reef Foundation.

**Competing interests:** The authors have declared that no competing interests exist.

proportional survival is acknowledged, we assume the existence of a relationship [8]. This growth time may impact the post-deployment survival rate, volume yield, or other factors [8, 9]. This proportional post-deployment survival rate determines the required production quantities to meet demand requirements. Planning the production of these cultivated resources requires a method which explicitly accounts for post-deployment survival as a function of growth time.

While the placement of facilities is fundamental to optimizing the logistics of delivery problems [10–12], few methods account for both growth time and facility size optimization. The optimal locations of service facilities to satisfy customer and production demands have been considered for decades [13]. In previous studies, formulations minimize the cost of meeting all demand with the facility location problem and maximize coverage due to insufficient resources with the maximal covering location problem [12, 14]. A variant of this problem addresses production demands by introducing modular capacities to size facilities [15, 16]. Other approaches use an integrated method of identifying the optimal location for facilities and incorporate facility capacity sizing decisions to satisfy demand, addressing location-dependent costs [17]. The effect of survival, dependent on in-process inventory times, and its effect on production quantity demands and facility sizing has received little attention and is especially relevant to coral aquaculture.

The worlds' coral reefs are in decline due to localized and global human and natural factors [18–21]. Given the large geographic scale (348,000 km2) and mass coral mortality over the Great Barrier Reef (GBR), large-scale reef restoration is being assessed [22, 23]. Coral aquaculture, as one of many restoration methods, is being considered to address mass coral mortality over the GBR [22]. Planning how to best use resources for aquaculture projects to cultivate coral for large-scale reef restoration is vital to efficiently and sustainably deploying grown corals [21]. Mathematical optimization is needed to determine the optimal number, size, and locations of aquaculture facilities to successfully fulfil these large-scale plans.

There are various approaches to using aquaculture to grow young corals to produce coral biomass for restoration. The nature of this planning problem is dependent on the specific style of aquaculture carried out. Corals can be propagated sexually through larval spawning or asexually by fragmentation. Whilst sexually propagated corals offer the potential to enhance genetic diversity of reefs, further considerations must be made to consider the timing and logistics of coral spawning [8]. Asexual propagation involves pre-settling fragmented corals onto deployment devices, and rearing corals until deployed. This allows greater control over production planning and scheduling. The placement of aquaculture facilities on land or in the sea is another important factor considered in these projects. *In-situ* approaches grow coral fragments or larvae in sea-based nurseries [23]. Land-based ex-situ aquaculture involves cultivating corals in a controlled environment on land before transporting and deploying young corals to host reef sites [22, 24]. This approach can lead to high quality control and throughput but requires establishment and operation of land-based facilities. Land-based aquaculture generates significant energy demands for pumping water and sustaining essential utilities. These energy demands, unless met by renewable sources, produce carbon emissions, contributing to climate change and reef degradation, reinforcing the degradation that the intervention is aiming to combat [25]. Operations need to be carefully assessed, optimized, and managed to address these issues.

In designing such facilities, the growth time allocated for corals after being fragmented and settled in facilities is considered. Optimizing growth time for large-scale coral aquaculture production is crucial to cost-effective production and deployment. The post-deployment survival rate drives production quantity requirements and facility sizing decisions. This production decision affects the optimization of facility locations and their respective sizing. By

characterizing the relationship between coral survival and coral growth time, a coral aquaculture program can be optimized. An optimal growth time accounts for the trade-off between minimizing facility costs (both capital and operational), whilst capitalising on the benefits of increased survival. Therefore, the space and energy savings of lower growth times compete with the benefits of reduced production quantity associated with larger growth times. The relationship between varying growth time and post-deployment survival is not well-defined in the literature reviewed. Coral survival rates are affected by their size at deployment [26–29]. Coral growth rates over time [24], estimates of age-dependent survival [6, 30–32], and the statistical significance of size- and age-dependence on survival [33] have been studied. The survival rate of juvenile sea cucumbers, settled on modular plates in tanks was assessed based on mean size [34]. This is described by similar processes and considerations to coral aquaculture in terms of an age-based survival relationship. The growth time of corals in a facility determines the age and size of corals deployed. However, survival rates based on coral growth time in a facility have not been well-defined in the literature, and this relationship is not considered in reviewed formulations.

Optimal cultivation timing needs to be integrated with facility location and sizing decisions to determine whether it affects optimal strategies to meet production demands. Existing models and approaches in the literature are not suited to solve this problem of locating and sizing facilities for coral aquaculture. To address this problem, we develop a new fully described integrated mathematical programming decision model that can tackle: (1) facility location, (2) facility sizing, and (3) a required production volume as a function of growth time. These three aspects have not been integrated in existing models. We consider coral survival rates to generate a minimum cost solution for the coral aquaculture production process by optimizing the facility location, sizing, and growth time simultaneously. The proposed formulation generates a facility location and sizing plan with an associated optimal growth time of corals in these aquaculture facilities to minimize production and transport costs. This study does not advocate for a specific aquaculture method, but rather optimizes the location and design of land-based aquaculture facilities in applications where they are being considered. We illustrate this model on a GBR case study and test the sensitivity of these parameters to quantify the value parameter certainty on the total cost and production planning decisions.

## 2 Methods

We propose a model that considers the construction and operation of land-based ex-situ coral aquaculture facilities over a prescribed planning horizon to cultivate and deploy corals to specified reefs in the wild. We then use this model to optimize the size and placement of the facilities to minimize amortized costs and meet demand for reef adaptation and restoration outcomes. Here we assume that planners have specified the use of land-based facilities rather than comparing different aquaculture strategies. Strategic and tactical decisions are considered to minimize costs by determining optimal facility location, size, and time periods to cultivate coral pre-deployment. The rearing time (referred to as growth time) for grown resources in a facility is explored as a key decision variable correlated to post-deployment survival. This growth time affects optimality as a driver of capital and operating costs and production throughput. We use a two-stage algorithm to generate a near-optimal solution to a mixed-integer nonlinear programming problem. This formulation is based on allocating demand for surviving resources at respective deployment locations to facilities based on their location availability. Production quantity dictates facility sizing, as quantities required to be produced are addressed based on the estimated proportion of cultivated resources that survive in the wild. Various deployment scenarios are assessed using given cost parameters, resource and

logistical demands and requirements, facility location availabilities, and survival considerations.

## 2.1 Model formulation

A mathematical model is required to characterize the relationship of survival with respect to growth time. We formulate a novel minimization problem as a mixed-integer nonlinear program. Given a set of demand location clusters $I$, with demand $d_i$ at each cluster $i \in I$, and set of candidate facility locations $J$, we aim to determine the optimal growth time $t^*$ and quantity of grown resource $X_{ij}$ to be produced in each facility $j \in J$ to be deployed at each demand cluster $i \in I$. Solutions provide recommendations for location, size, and number of facilities. An optimal solution is obtained by following the algorithm described in section 2.2, and minimizing the total cost, represented mathematically by our objective function value in section 2.1.4, Eq (1). Problem instances are specified by deployment demand, cost parameters, survival relationships, and candidate facility sites.

The proposed model considers facility costs based on floor area requirements, variable costs incurred for cultivating resources, and transport costs for deploying these resources from their growing facility to deployment sites. An optimal growth time and location-allocation plan of resources grown in various facilities is generated to meet given deployment demands in respective demand locations. Estimates of survival rates over a range of growth times inform deployment frequency decisions. Demand generated from spatial data is given based on external assessments and decisions. Therefore, a deterministic model is implemented and constrained to a set of identified demand points. Grown resources are deployed to these sites to satisfy specified demand quantities. Demand at each site is assigned to a facility, and a growth time determined to minimize total costs.

**2.1.1 Decision variables.** Decision variables are introduced to determine the optimal deployment planning decisions. These variables dictate which facilities are opened ($Y_j$), the quantity of grown resource produced at each facility assigned to each demand cluster ($X_{i,j}$), and the growth time to hold the resource in a facility ($t$). A decision variable ($T_{i,j}$) is introduced to specify the number of trips required from facility $j$ to demand cluster $i$, based on the production quantity $X_{i,j}$ and capacity constraints of transport vessels $s_c$–outlined below in section 2.1.4 and Eq (2).

**2.1.2 Survival function.** Some proportion of resources grown in a facility will survive in the wild. For this model, we assume the proportional survival of each grown resource depends on the amount of time it has been grown in a facility. This growth time in a facility can significantly affect the cost effectiveness of the solution. We assume the existence of a known function, $P(t)$, to calculate the proportion of grown resources that survive as a function of growth time $t$, and explore changes to this relationship. For a given total demand $D$ and production quantity $X$, it is then required that $P(t) \cdot X = D$. Later we describe and explore several potential functional forms for $P(t)$. The proportional survival of coral is constrained by a minimum and maximum rate ($p_{min}$ and $p_{max}$) that define the asymptotes and limits of the survival functions. A variable ($p_{min_{allow}}$) is used to truncate proportional survival values below an allowable limit to eliminate calculations with very small survival rates, and therefore very large associated production quantities, causing large computation times.

**2.1.3 Costs.** The proposed formulation aims to optimize facility sizing and location decisions, given associated costs. These costs consist of capital and operational costs. The fixed and variable (per m$^2$) capital costs ($c_f^{cap}$ and $c_v^{cap}$ respectively) associated with building and opening a facility, as well as variable costs ($c_f^{op}$) of operating a facility, are assumed to be homogenous across every candidate location $j \in J$ for the purposes of this study. Capital costs are

**Table 1. Description of model parameters and assumed values based on expert elicitation and estimates.**

| Notation | Case Study Value | Description |
|---|---|---|
| $d_c$ | 1 | Coral density–corals required per m$^2$ of area restored |
| $r_{width}$ | 20 | Width of restoration atoll area to be restored (m) |
| $r_r$ | $\sqrt{\pi \, a_r}$ | Radius parameter for reef $r \in R$ in dataset calculated from area |
| $d_r$ | $c_d \begin{cases} \pi(r_r + {}^{r_{width}}/_2)^2 & \text{if } r_r \leq 100 \\ \pi\left[(r_r + {}^{r_{width}}/_2)^2 - (r_r + {}^{r_{width}}/_2)^2\right] \text{else} \end{cases}$ | Demand for number of corals to service reefs $r \in R$ for each cluster per year |
| $P(t)$ | $\begin{cases} 0 & \text{if } p_{max} - \dfrac{c}{(t-t_0) + \dfrac{c}{p_{max}}} \leq 0 \\[3em] p_{max} - \dfrac{c}{(t-t_0) + \dfrac{c}{p_{max}}} & \\[3em] p_{max} & \text{if } p_{max} - \dfrac{c}{(t-t_0) + \dfrac{c}{p_{max}}} > p_{max} \end{cases}$ | Proportion of surviving corals deployed at reef $i \in I$ as a function of residence time. |
| $c_f^{cap}$ | \$10,000,000 | Fixed capital cost (\$) for the opening of each facility. This is amortized at 5% p.a. over 25 years. |
| $c_v^{cap}$ | \$45,000 | Variable capital expenditure to build facility (\$ per m$^2$). This is amortized at 5% p.a. over 25 years. |
| $s_c$ | 432,000 | Maximum capacity of a deployment vessel, measured in number of coral units. |
| $a$ | $\frac{8.05 \text{ m}^2}{24000 \text{ units}} \approx 3.35 \text{cm}^2/\text{coral unit}$ | Area required (m$^2$) for each coral unit. |
| $c_f^{op}$ | $\frac{\$1.92/3 \text{ units}}{0.25 \text{ years}} \approx \$2.56/\text{unit/year}$ | Cost (\$) per year for coral unit to reside in a facility. |
| $c_d^{op}$ | \$100/km | Cost (\$) to transport each coral unit from facility $j \in J$ to reef $i \in I$. |

amortized over the prescribed life-of-asset period and with an assumed periodic interest rate. The cost parametrization is outlined in Table 1. These cost estimates are based on expert elicitation and estimates for facilities that have not yet been built. As a result, these estimates have high associated uncertainty and should be refined if plans are being developed.

The required size (i.e., gross floor area) of each facility is determined by its production capacity and the coral growth time. The floor area, $a$, required to house each unit of grown resource is calculated by dividing the floorspace required for each enclosure unit by the number of resources grown in that unit to return an area parameter per grown unit. By Little's law [35], the total area required for production of grown resources for each demand cluster $i \in I$ at facility location $j \in J$ can be calculated as $a \times t \times X_{i,j}$. This total area is then multiplied by the variable capital facility cost, $c_v^{cap}$, to obtain the cost per year of growing resources in a facility.

Operational expenditure consists of facility and deployment costs. Facility operating costs, $c_f^{op}$, are measured as cost per unit of grown resource per year and are consistent across facilities. These facility costs, $c_f^{op}$, represent the cost per unit of grown resource in production within the facility per year, and are consistent across facilities. Deployment costs, $c_{i,j}^T$, correspond to transportation, accounting for the use of vessels. Geospatial cluster coordinates are used to generate a distance matrix from each facility $j \in J$ to each demand cluster $i \in I$. An additional intra-cluster distance term is included based on the number of demand points in the cluster. The transport cost is obtained by multiplying these distances by $c_d^{op}$, a fixed cost per kilometre travelled to obtain the cost matrix, $c_{i,j}^T$, representing the return trip cost values for each trip, accounting for facility-cluster pairs and reef deployments.

**2.1.4 Objective function and constraints.** Our objective is to minimize the total cost of meeting production demands over the life of the production program. The objective function, Eq (1), aims to minimize the total (fixed and variable, capital and operating) costs of facilities accounting for the number of resources required to be grown, their growth time and transport

costs. Constraint (2) ensures that the number of trips from each facility to each demand cluster is sufficient to transport the number of resources required to meet the demands of that location, considering the capacity of the transport vessel. Constraint (3) ensures that for each facility, the total number of trips deployed per year is at least the number of deployment events. Constraint (4) ensures that the quantity of resource grown that survives post-deployment meets the requirements for each demand location. Constraint (5) ensures the variable $Y_j$ must be non-zero if there is non-zero production at facility location $j$, where $M$ is the total demand divided by the minimum (non-zero) survival rate, $P(t)$, for $t$ within the range $[t_{min}, t_{max}]$. Domains of variables are given in (6)-(9). The growth time, $t$, is assessed at increments of 0.02 years ($\Delta t$) between $t_{min}$ and $t_{max}$.

$$\text{Minimize} \quad \sum_{j \in J} \left\{ c_f^{cap} Y_j + \sum_{i \in I} [c_{i,j}^T T_{i,j} + X_{i,j} P(t)(c_v^{cap} a + c_f^{op})] \right\} \tag{1}$$

Subject to:

$$X_{i,j} \leq s_c T_{i,j} \forall i \in I, j \in J \tag{2}$$

$$t \cdot \sum_{i \in I} T_{i,j} \geq 1 \forall j \in J \tag{3}$$

$$P(t_{grow}) \sum_{j \in J} X_{i,j} \geq d_i \forall i \in I \tag{4}$$

$$\sum_{i \in I} X_{i,j} \leq M Y_j \forall j \in J \tag{5}$$

$$X_{i,j} \geq 0 \forall i \in I, j \in J \tag{6}$$

$$T_{i,j} \in \mathbb{Z}^+ \forall i \in I, j \in J \tag{7}$$

$$Y_j \in \{0, 1\} \qquad\qquad \forall j \in J \tag{8}$$

$$t_{min} \leq t \leq t_{max} \tag{9}$$

## 2.2 Solution method

As it is formulated, (1)–(9) is a mixed-integer nonlinear programming problem. This is not ideal for the application of a standard solver and verifying optimality of solutions. However, if the value of the variable $t$ is fixed, the model is linear and relatively straight forward to solve. Thus, a nested two-stage approach is proposed with the outer stage systematically perturbing $t$, which is fixed, and the resulting MIP solved as the inner stage. The algorithm is given below.

```
Algorithm: Solve facility location and sizing problem
Inputs:
    𝔉 Problem instance parameter set
    Δt Step size for perturbing growth time
Outputs:
    t* Optimal growth time (within ±½ Δt)
    𝔛* Optimal set of values of other decision variables given t^*
z* Optimal objective function value
Procedure: Solve facility location and sizing problem
    Let z* = ∞
    For t̂ ∈ {t_min, t_min + Δt, t_min + 2Δt, ..., t_max}
        (𝔛, z) ← solve MIP (1)-(9) with parameter set (𝔉, t̂)
```

```
          If z<z* then
              Let (𝔛*, t*, z*) = (𝔛, t̂, z)
          End if
      End for
End procedure
```

## 2.3 Case study

We assess a large-scale coral aquaculture program aiming to meet the restoration demands of reefs in the GBR Marine Park at minimal cost. A proprietary dataset containing the name, latitude, longitude, and area for 2816 reefs in the GBR was provided by The Australian Institute of Marine Science (AIMS). The restoration needs and objectives of managers will affect the reefs that require treatment, therefore a randomly selected subset of 50 reefs from the 2816 candidate reefs is considered to generate demand for coral at specific locations. Reefs represent locations with demand for coral fragments, a resource grown in onshore facilities at a coastal port. Reefs are clustered to reduce distances travelled by allowing reefs to be grouped based on the capacity availability of deployment vessels, and the relative location of target reefs (explained in section 2.4.2). As seen in Fig 1, reefs are scattered along the length of the coast from the Northern peninsula.

Asexual coral propagation is considered for coral production in land-based, ex-situ coral aquaculture facilities. Corals are to be deployed to selected reefs to satisfy a given demand–a given number of deployed corals that must survive at each reef one year post-deployment. The estimated survival rate is critical to the accuracy of results, however the relationship between coral survival probability and growth time in a facility have not previously been well-defined. These ex-situ aquaculture facilities contain aquarium tanks where coral fragments are grown on settlements devices. These devices are then transported from facilities to ports on trucks, and from ports to reefs via boats. The deployed devices are then systematically released at determined reef locations. This deployment process is repeated periodically, based on the aquaculture facility schedule, servicing reefs over the course of the year. For this model, facilities are assumed to be located at a port, which removes the need for trucks in the supply chain and creates a single mode of transport–directly from facility to reef via boat.

We parameterize capital (fixed and variable) expenditure, and (variable) operational costs of facilities, and transport costs for corals from facilities to reefs from consulting reports commissioned by the partner organisations and data provided to the research team by AIMS (see Table 1).

## 2.4 Case study parameterisation

A base case scenario is established by a premise of informed assumptions and estimates to generate an initial result as a comparison benchmark. Potential aquaculture facility candidate locations are seven coastal cities in Queensland, Australia: Bundaberg, Gladstone, Rockhampton, Mackay, Airlie Beach, Townsville, and Cairns. The operational and financial figures considered are informed by commissioned consulting reports provided by AIMS. These provide a relative weighting of different costs, and floorspace constraints to inform an estimate. We assume that building these facilities includes \$10,000,000 fixed capital, and \$45,000/m$^2$ variable capital costs. Capital costs are amortized over 25 years with a periodic interest rate of 5% p.a. Operationally, coral fragments are to be settled on a matrix of settlement tiles; each square of tile is considered a unit. The facilities incur a yearly expenditure of \$2.56/unit. The coral aquaculture production floorspace capacity is limited by settlement tanks that hold 24,000 coral units per 8.05m$^2$, inclusive of an allowance for walkways. The cost of deployment is \$100/km travelled by ship, with a homogenous fleet of vessels, each with a capacity of 432,000 coral units. These parameter values, as well as all other values and calculations for parameters are outlined in Table 1.

**2.4.1 Base coral survival.** For the base case, we assume the survival function is an increasing concave curve that approaches a steady-state asymptote. This function is defined by scaling factor ($c$) to determine the significance of the curve with respect to changes in growth time, and a horizontal shift factor ($t_0$) that shifts the curve horizontally, signifying the x-intercept. The proportional survival is restricted to positive values less than the maximum proportional survival ($p_{max}$), the asymptotic limit of the function. The initial assumptions for base case parameter values are ($c, t_0, p_{max}$) = (25, 0, 0.1), and the function is mathematically represented as

$$P(t) = \begin{cases} 0 & \text{if } p_{max} - \dfrac{1}{c(t - t_0) + 1/p_{max}} \leq 0 \\[2ex] p_{max} & \text{if } p_{max} - \dfrac{1}{c(t - t_0) + 1p_{max}} > p_{max} \\[2ex] p_{max} - \dfrac{1}{c(t - t_0) + 1p_{max}} & \text{otherwise} \end{cases} \tag{10}$$

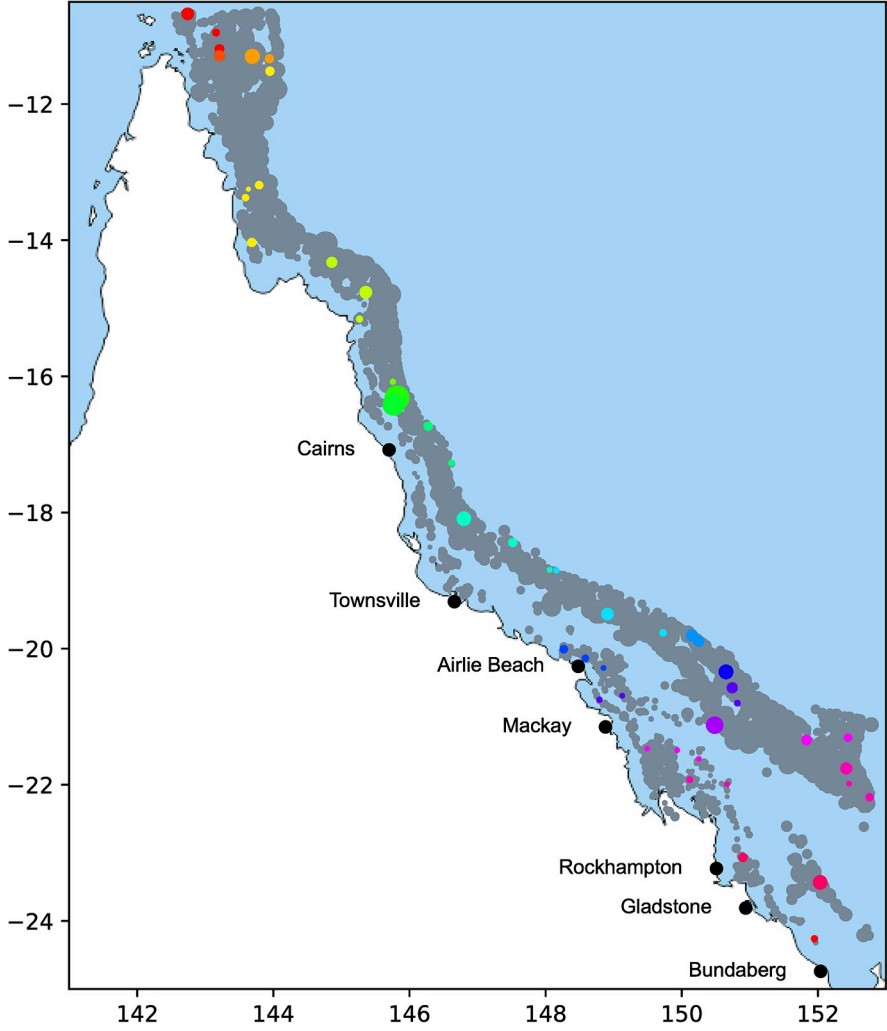

**Fig 1. Reefs in AIMS dataset dispersed along the coast of Queensland.** Total set of 2816 reefs are size-scaled by area. A subset of the 50 randomly chosen reefs used for the case study are coloured by cluster, with the remaining reefs coloured grey. Port locations are coloured black. Reprinted from [36] under a CC BY license, with permission from d-maps.com, original copyright 2021.

**2.4.2 Reef demand and clustering.** The demand, $d_r$, for corals at each reef $r \in R$ is a function of the area of restoration for that reef. This restoration area is characterized by an atoll, geometrically represented as a circular area, around the circumference of a reef with a given width, $r_{width}$. A formula for calculation of these values is given in Table 1. Reefs are clustered to optimize transport logistics in the deployment of corals. Clusters, denoted as $i \in I$, are formed by sweeping from North to South through the subset of reefs. Reefs are clustered if adequate ship capacity is available to service the Southern adjacent reef. The geospatial centroid is then used as the coordinates of the cluster, with an allowance for additional intra-cluster transport based on the number of reefs in the cluster.

## 2.5 Sensitivity analysis

At this point in time, coral aquaculture deployment for the GBR is still in research and development, so many of the case study parameters are estimated or not known. An extensive sensitivity analysis is conducted to illuminate the key factors impacting FLSCA solutions across a range of diverse problem instances and highlight the value of obtaining parameter certainty. The problem instances investigated are generated by independently perturbing specific parameters of the base case parametrization. These perturbed parameters include the selection of reefs serviced, the magnitude of cost parameters, the maximum proportional survival rate of corals post-deployment, and the functional form and associated parameters of the coral survival curve over varying growth times in a facility.

**2.5.1 Reef selection.** We assume a specific subset of reefs to be targeted for coral aquaculture deployment is provided when performing the optimization. Here, we randomly generate these, however in a real application these are likely to be provided. To investigate solution sensitivity to reef selection, 100 problem instances each consisting of 50 reefs are randomly generated. The solutions are assessed in terms of optimal facility number and location to service these reefs.

**2.5.2 Cost parameters.** Optimal total cost, facility number and locations, and growth time are assessed based on cost parameter multiplication by a scaling factor raised to the power of two. Positive and negative integers are considered, raised to the power of two, to scale cost parameters with respect to base case values. Scenarios are recorded until a threshold is passed which changes the optimal number and/or location of facilities. The impact of this manipulation is assessed with respect to each cost parameter independently.

**2.5.3 Maximum proportional survival.** The maximum proportional survival rate is an upper limit for the proportion of corals that survive post-deployment. This limit is assessed from 10% to 100% at increments of 10%. The asymptotic distribution, utilized in the base case scenario, is used in the sensitivity of maximum proportional survival. The total cost, growth time, and number and location of facilities for optimal solutions is used to assess the sensitivity of maximum proportional survival.

**2.5.4 Coral survival functional form.** Due to uncertainty in the functional form of coral survival, we consider a range of functional forms following different trajectories over the range of growth times. Four functional forms (including the asymptotic function as the base case) are considered to relate proportional survival of corals with respect to growth time in a facility (Fig 2). Each of these functions are defined by two parameters, scaling factor ($c$), and horizontal shift ($t_0$), varied to assess solutions and their sensitivity.

The first alternative functional form (to the base case, Eq (10) in Section 2.4.1) is Eq (11), a linearly increasing function with a maximum proportional survival. The scaling factor ($c$) represents the gradient, and horizontal shift ($t_0$) is the x-intercept for the linear function. The

linear relationship of proportional coral survival with respect to growth time is

$$P(t) = c(\text{t} - t_0).$$ (11)

The next functional form is represented by the logistic function, a sigmoidal curve bound by the maximum proportional survival asymptotic limit, seen in Eq (12). The characteristic of this curve is a convex curve at low growth times, reaching an inflection point, and concave curve at growth times greater than the inflection point. The scaling factor ($c$) is correlated to the gradient, and horizontal shift ($t_0$) correlates to the inflection point for the logistic function. Here,

$$P(t) = \frac{p_{max}}{1 + e^{-c(t-t_0)}}.$$ (12)

Finally, the proportional survival is modelled in Eq (13), as a function that reaches a peak before decreasing towards a horizontal asymptote. This reflects the characteristics of corals developing a reliance on the artificial conditions resulting in a lower resilience in the wild when kept too long in a facility. The Eq is based on a gamma distribution constrained to curves where $\alpha = \beta$ [37]. We refer to this as a pseudo-gamma function because it is based on one instance of this gamma distribution, without any relative variance of these $\alpha, \beta$ parameters. For this function, the scaling factor ($c$) is correlated to the gradient of the peak, and the horizontal shift ($t_0$) to the x-intercept. The vertex point is characterized by a growth time of $^1/_c + t_0$. This functional form is expressed as

$$P(t) = c(\text{t} - t_0)e^{-c(\text{t}-t_0)}.$$ (13)

Each of the functional forms are constrained to positive values less than the maximum proportional survival rate ($p_{max}$, represented by the black dashed line in Fig 2). This defines a range of proportional survival for corals that survive post-deployment. Proportional survival rates outside the given range are truncated. The constraint applied to each of these functions is

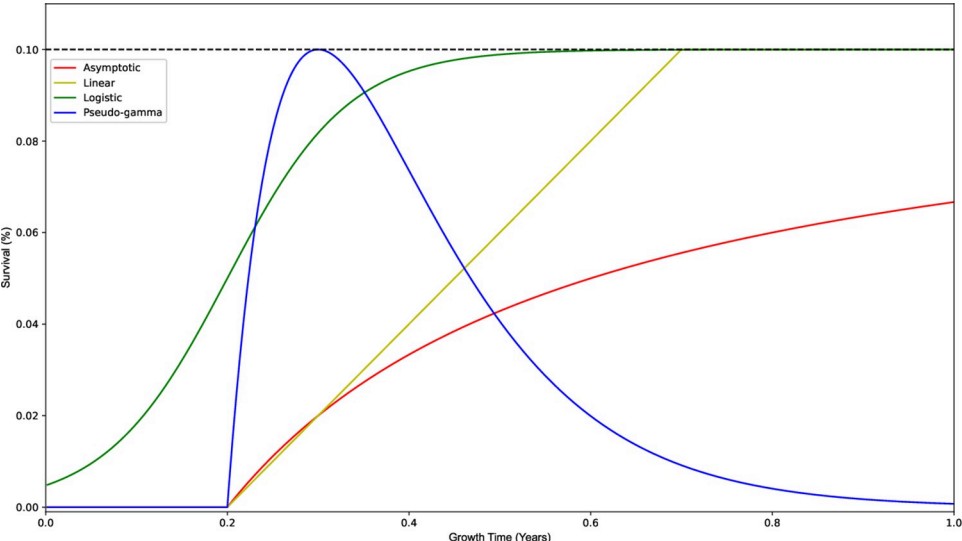

**Fig 2. Proportional survival plot of asymptotic, linear, logistic, and pseudo-gamma functional forms against growth time.** $c$ = 25, 0.2, 15, 10 respectively. $p_{min} = 0$, $p_{max} = 0.1$ (black dashed line), $t_0 = 0.2$.

characterized by

$$P(t) = \begin{cases} 0 & \text{if } P(t) \leq 0 \\ p_{max} & \text{if } P(t) > p_{max} \,. \\ P(t) & \text{otherwise} \end{cases} \tag{14}$$

The functional forms have differing characteristics and respective parameter variance ranges. The horizontal shift ($t_0$) is varied consistently between 0 and 0.5 across all functions. The range of the gradient term ($c$) is varied between the limits of 5 and 55, 0 and 0.5, 5 and 55, and 5 and 30, respectively, for the asymptotic, linear, logistic, and pseudo-gamma functional forms.

### 2.6 Computational configuration

The inner problem (MIP) is solved using the OR-Tools package [38] for Python, by applying its default MIP solver. This is run on an Intel Core i7 with a 1.90GHz processor and 16GB RAM. Computation times stated are based on this configuration.

## 3 Results

### 3.1 Case study

The base case optimal solution includes establishment of five facilities, in Gladstone, Rockhampton, Mackay, Airlie Beach, Townsville and Cairns (Fig 3) to service 50 reefs (Fig 1) in 20 clusters. This optimal minimum cost result returns a total of $172.5 million consisting of 53% operational facility costs, 23% transport, 22% variable capital, and 2% fixed capital costs. Based on the asymptotic survival function used, the optimal growth time is 0.10 years, with a survival rate of 2% thus producing 356.2 million corals to meet the demand of 7.124 million corals surviving one year post-deployment. The CPU time for solving this problem instance was 2 minutes ± 20 seconds.

### 3.2 Sensitivity analysis

**3.2.1 Reef subset.** The frequency of facilities appearing in optimal solutions over different reef subsets is shown in Fig 4. Computational run times for each problem instance was in the order of minutes. There is significant confidence (100% of solutions) in Cairns as an optimal facility location, and substantial confidence in Rockhampton and Airlie Beach (95% and 80% respectively), as well as Mackay and Townsville (73 and 55% of solutions respectively) as assessed across 100 reef subsets. Gladstone and Bundaberg appeared in 2 and 1% of optimal solutions respectively (Fig 4). This provides strong confidence in the establishment of a facility in Cairns. Further analysis was undertaken to provide greater insight into optimal locations for additional facilities.

The optimal number of facilities to meet demand varied across problem instances. Optimal solutions contained various combinations of facility locations (Fig 5), with most optimal solutions containing four facilities (55%). A substantial number of optimal solutions contained five or three facilities (24 and 20% respectively), and very few optimal solutions contained six facilities (1%).

The results in Fig 5 show that as the optimal number of facilities increase, the variance decreases between frequency of port locations. Five and six facility solutions (green and blue bars in Fig 5) have no significant difference in frequency across facility locations. However, variance is greater in three and four facility solutions (red and yellow bars in Fig 5).

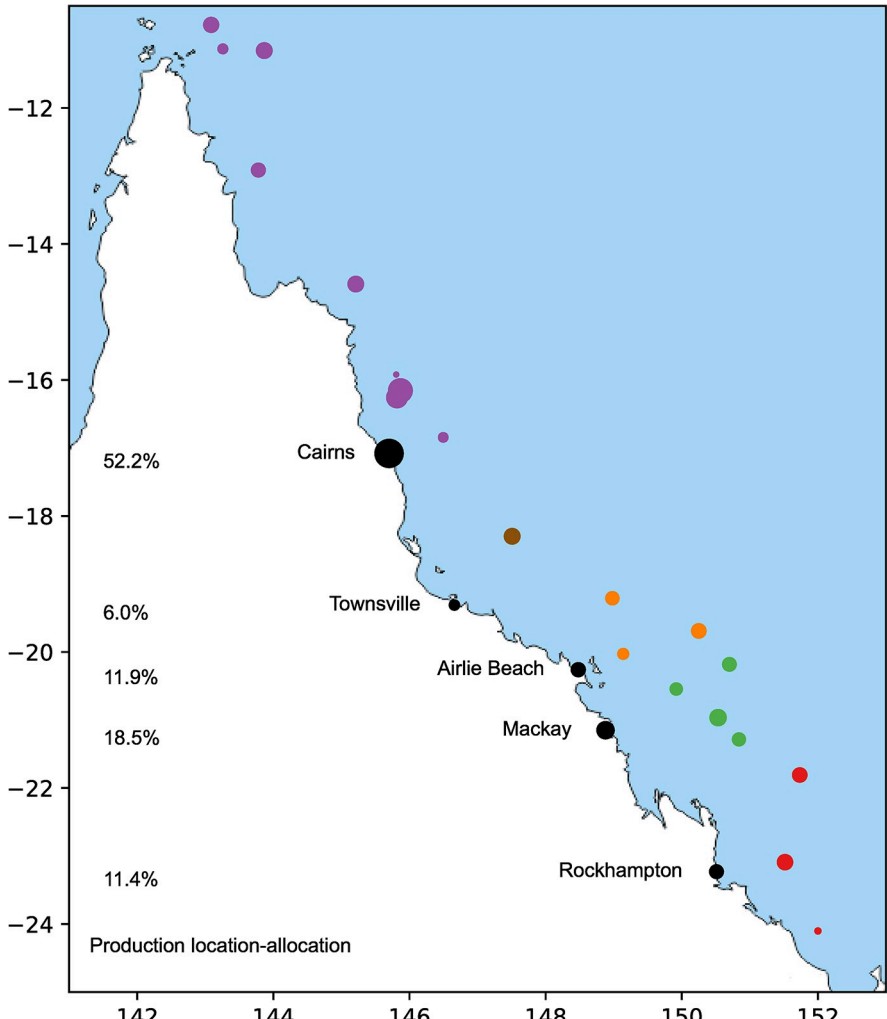

**Fig 3. A spatial plot of optimal solution for parametrized base case.** Reef clusters are sized by demand quantity and coloured by optimal port assignment–Cairns in purple, Townsville in brown, Airlie Beach in orange, Mackay in green, and Rockhampton in red. Optimal facility locations are coloured black sized by their production quantity, and indicated as proportion of total coral produced. Reprinted from [36] under a CC BY license, with permission from d-maps.com, original copyright 2021.

A trend of optimal facility location (based on the maritime logistics) is apparent within each series of $n$ facility solution. Three facility solutions favour Cairns, Airlie Beach, and Rockhampton (red bars in Fig 5); four optimal facilities add Mackay into this subset (yellow bars in Fig 5); five optimal facilities include Townsville (green bars in Fig 5); and six optimal facilities utilize Bundaberg (not shown in Fig 5). Each of these series grows, without conflicting or contradicting another. Bundaberg and Gladstone are not favoured as optimal facility locations (appearing in 1 and 2% of optimal solutions respectively, see Fig 4) and are therefore not included in Fig 5. The analysis of reef selection highlights the impact of the reef subset on the optimal facility number and placement, as well as the inference of optimal facility locations with knowledge of the number of facilities established.

**3.2.2 Cost parameters.** The sensitivity of each respective cost parameter is assessed incrementally. Variance in the relative weighting of cost parameters affects minimum cost solutions and the optimal number, location, and size of facilities. Computation time for running this

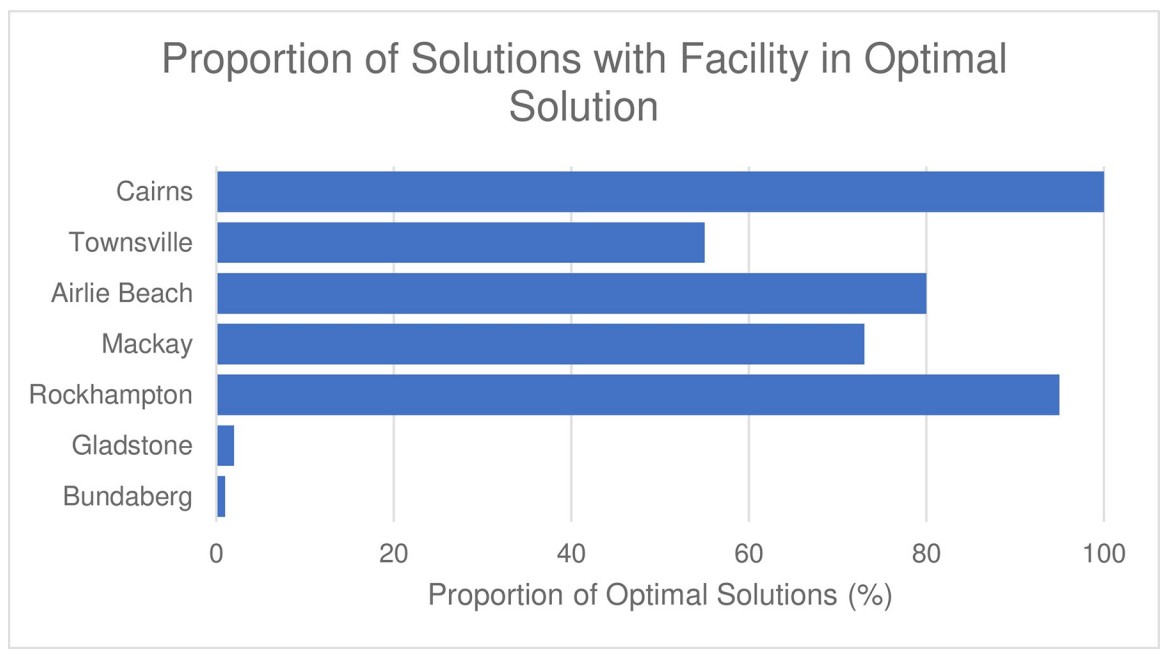

**Fig 4. A proportional frequency histogram of each potential facility location appearing in optimal solutions based on 100 random subsets of 50 reefs.**

analysis was 1 hour, 24 minutes, 54 seconds. Cost parameter sensitivity results can be seen in Table 2.

The scaling of fixed facility capital costs had little impact on optimal growth time and associated production quantity. However, scaling of this fixed cost is inversely correlated to the optimal number of facilities. When fixed capital cost is scaled by $2^1$, three facilities are optimal

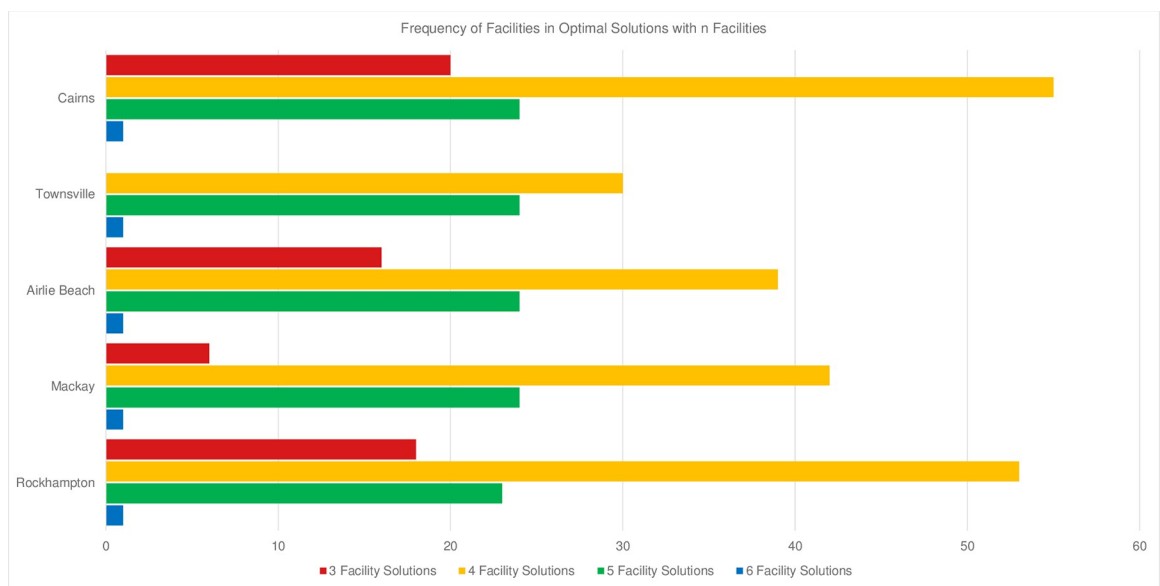

**Fig 5. Frequency of potential facility locations included in optimal solutions for 100 random subsets of 50 clustered reefs.** Bars are grouped by facility location and coloured by the number of optimal facilities. This plot excludes Bundaberg and Gladstone which appear in less than 5% of optimal solutions.

**Table 2. Cost parameter sensitivity analysis.** The base case scenario is listed, as well as solutions with cost parameter scaling variations resulting in optimal solutions with a different number of optimal facilities to the base case scenario. Facility locations are Bundaberg (B), Gladstone (G), Rockhampton (R), Mackay (M), Airlie Beach (A), Townsville (T), and Cairns (C).

| Cost Parameter | Scaling Factor ($2^x$) | No. Optimal Facilities | Facility Locations | Optimal Residence Time (years) | Optimal Production Quantity (million) | Optimal Total Cost ($ million) |
|---|---|---|---|---|---|---|
| Base Case | 0 | 5 | R, M, A, T, C | 0.10 | 356.2 | 172.5 |
| Facility—Fixed Capital | 1 | 3 | R, M, C | 0.12 | 308.7 | 175.6 |
| | -2 | 6 | G, R, M, A, T, C | 0.10 | 356.2 | 169.8 |
| Facility—Variable Capital | 10 | 6 | G, R, M, A, T, C | 0.10 | 356.2 | 39,200.0 |
| | -inf | 5 | R, M, A, T, C | 0.14 | 274.8 | 132.8 |
| Facility—Variable Operational | 9 | 6 | G, R, M, A, T, C | 0.10 | 356.2 | 46,770.0 |
| | -inf | 5 | R, M, A, T, C | 0.20 | 213.7 | 73.2 |
| Deployment—Variable Operational | 3 | 6 | G, R, M, A, T, C | 0.34 | 155.1 | 333.8 |
| | -1 | 3 | R, M, C | 0.10 | 356.2 | 152.6 |

(in Rockhampton, Mackay, and Cairns), and when scaled by $2^{-2}$, six facilities are optimal (adding Gladstone to the base case solution).

The scaling of all variable cost parameters is positively correlated to the optimal number of facilities. An increase in these parameters increases the number of optimal facilities due to the inherent relative decrease in weighting of fixed facility capital costs. The optimal growth time is inversely correlated to scaling of variable capital and variable operational facility costs, whilst positively correlated to variable deployment costs. The optimal number of facilities and locations are less sensitive to changes in variable capital and variable operational facility cost parameters, with larger scaling factors ($2^{10}$ and $2^9$) required to influence change than for other cost parameter changes. Variable facility capital cost increased by less than $2^{11}$ and variable facility operational cost increased by less than $2^{10}$ returned solutions with optimal facility locations consistent with the base case. Variable capital and operational facility costs are incurred for each unit of facility area required and therefore coral produced. With no change in production quantity, large scaling of these parameters results in large total costs ($39.2 and $46.8 billion respectively), two orders of magnitude larger than all other scenarios. Scaling variable operational deployment costs by $2^3$ resulted in an additional optimal facility and a significantly increased optimal growth time (0.34 years); whilst halving operational deployment resulted in an optimal solution with three facilities and an optimal growth time consistent with the base case. The scaling of deployment cost drives the significance of deployment transport distances. Additional facilities (closer to reefs) reduce distance travelled. As the scale of deployment cost is increased (relative to facility fixed capital expenditure), additional facilities begin to reduce total cost.

**3.2.3 Maximum proportional survival.** The impact of an incrementally varied maximum proportional survival rate on the optimal number of facilities, total cost, and growth time is assessed (see Table 3). The optimal total cost has an exponentially decaying trend (orange line in Fig 6) with respect to increasing maximum proportional survival. The optimal number of facilities (grey bars in Fig 6) are reduced incrementally from five facilities at 10% maximum proportional survival to two facilities at a maximum of proportional survival of 50% and above. From the results obtained, there is no clear correlation between optimal growth time with respect to changing the maximum proportional survival rate (blue line in Fig 6). Computation time for running this analysis was 6 minutes, 53 seconds.

**Table 3. Maximum proportional survival variance sensitivity.** Sensitivity analysis results of maximum proportional survival rate on asymptotic survival function servicing base case subset of 50 reefs. The total cost, facility number and location, residence time, and production quantity are displayed for each. Locations are represented by the first letter of the names: Bundaberg (B), Gladstone (G), Rockhampton (R), Mackay (M), Airlie Beach (A), Townsville (T), and Cairns (C).

| Maximum Proportional Survival Rate (%) | Total Cost ($ million) | Facilities | Locations | Residence Time | Production Quantity (million) |
|---|---|---|---|---|---|
| 10 | 170.100 | 6 | G, R, M, A, T, C | 0.10 | 356.2 |
| 20 | 53.190 | 4 | G, M, A, C | 0.08 | 124.7 |
| 30 | 29.070 | 3 | G, M, C | 0.06 | 76.52 |
| 40 | 19.540 | 3 | G, M, C | 0.06 | 47.49 |
| 50 | 14.990 | 3 | G, M, C | 0.06 | 33.25 |
| 60 | 12.310 | 3 | G, M, C | 0.06 | 25.07 |
| 70 | 10.610 | 2 | M, C | 0.08 | 17.45 |
| 80 | 9.296 | 2 | M, C | 0.06 | 16.33 |
| 90 | 8.497 | 2 | M, C | 0.06 | 13.78 |
| 100 | 7.771 | 2 | M, C | 0.06 | 11.87 |

**3.2.4 Coral survival functional form.** The total cost, growth time, and number of facilities in optimal solutions are affected by changes in the scaling factor and horizontal shift across various coral survival functions (Fig 7). The total computation time for running this analysis across eleven increments for two parameters of four survival functions was 5 hours, 18 minutes, 15 seconds.

The horizontal shift is directly correlated to the time required for corals to be settled in a facility to reach maximum proportional survival across all survival functions; whilst the scaling factor varies in effect across functional forms, as expressed in respective coral survival functional forms (see section 2.5.4). The lowest total cost across all solutions ($21.97 million) corresponds to the lower bound growth time (0.02 years), whilst the largest total cost ($533.33 million) corresponds to the largest growth time growth time (1.00 years).

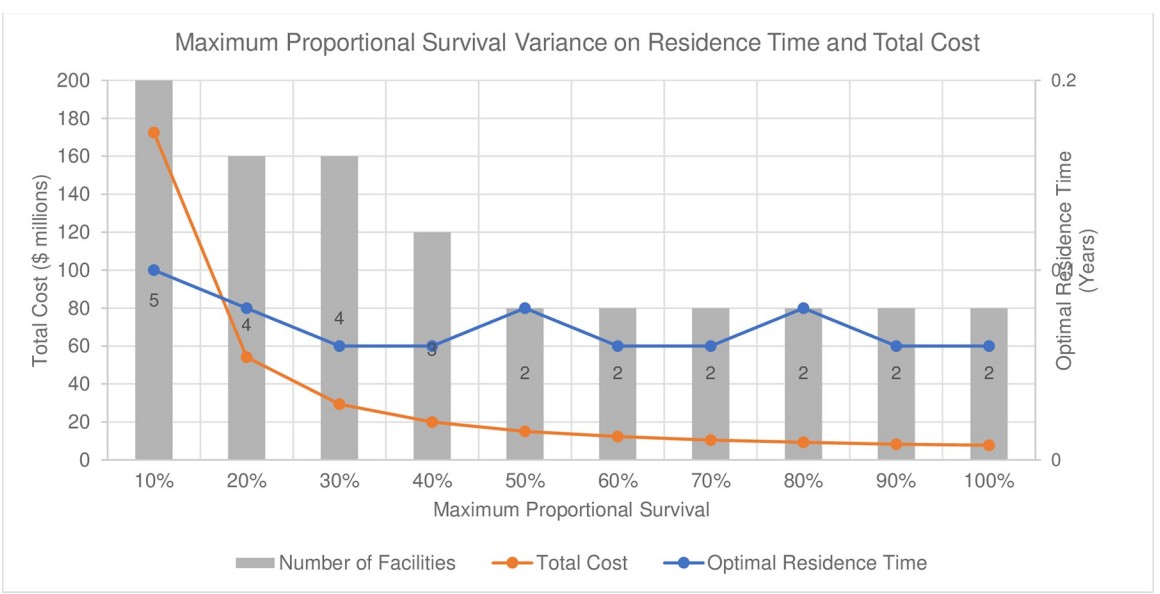

**Fig 6. Maximum proportional survival variance.** Optimal number of facilities (numbered in bars), total cost ($ million), and optimal growth time (years) relative to change in the maximum proportional survival limit for the base asymptotic survival function ($c = 25$, $t_0 = 0$).

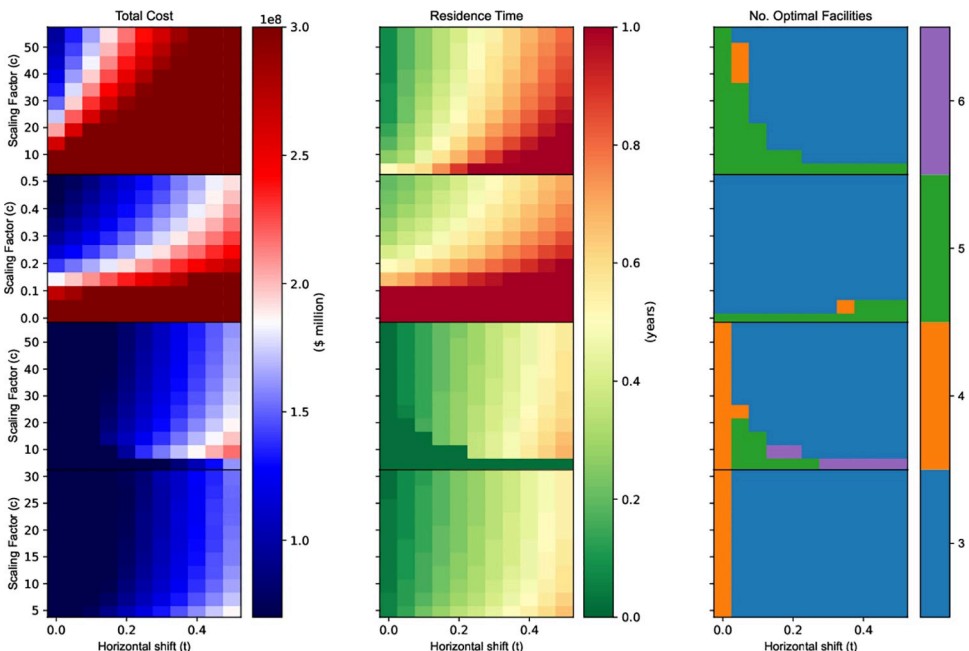

**Fig 7. Impact of change in gradient (*c*) and horizontal shift (*t₀*) on optimal solutions for different survival curve relationships.** The asymptotic, linear, logistic, and pseudo-gamma distributions are plotted in panels a, b, c; d, e, f; g, h, i; and j, k, l; respectively. Column 1 (panels a, d, g, j) show total cost of the optimal solution, represented by the objective value from $70 million to $300 million. Column 2 (panels b, e, h, k) show growth time (years) from 0 to 1 year in 0.02 increments. Column 3 (panels c, f, i, l) show optimal number of facilities from 3 to 6.

Across all survival functions, the lowest total cost values and optimal growth times are obtained with combined low horizontal shift ($t_0$) and high scaling factor ($c$) values (Fig 7). However, whilst the minimum optimal total cost by the logistic survival function (Fig 7G and 7H) is consistent with this trend; a grouping of low costs and corresponding low growth times are found with a low scaling factor. The results produced by low logistic scaling factors is caused by a weakened survival function gradient as the scaling factor approaches zero, producing results with rounding errors, see Eq (12).

## 4 Discussion

We have demonstrated a method to inform decisions regarding optimally sizing and locating production facilities and grow times for grown resources based on their survival rates. Our model provides a methodological framework to optimize the production of grown resources in facilities to meet spatially dispersed deployment demands. We parameterized and solved this problem in relation to growing coral and deploying them to candidate reefs for a GBR case study. We used potential facility locations at the main ports along the coast of Queensland, an AIMS reef database and monetized the solution by quantifying homogeneous fixed and variable facility costs, variable operational facility costs, and deployment costs. It is noted that the restoration case study considered is an expensive endeavour, and the magnitude of costs are within an expected range considering the given scale. Our method is relevant and transferrable to other datasets with differing demand points, production facility locations, and cost parameters. The most directly transferrable application would be grown resources with associated survival rates, as the production quantity and consequent facility sizing have been formulated to suit this scenario. The model could be easily adapted to suit the production of

other grown resources whose value is dependent on growth time through other proxies, including volume yield or product quality.

The facility location decisions are particularly important as they signify long-term strategic planning, considering the large associated fixed capital costs and significant costs associated with sub-optimal facility establishment. Whilst these decisions are strongly dependent on the subset of reefs selected; Cairns, the northernmost candidate location, has shown to be critical to addressing demand in the far northern GBR region, included in 100% of optimal solutions (Fig 4). The optimal location is largely driven by the location of Cairns being the most northern substantive port in the GBR, and around half of the reefs in the GBR lie to the north of Cairns. This confidence in Cairns as a suitable facility location is based on random subsets of reefs. However, the selection of reefs may not be random. Confidence of Cairns as an optimal facility location may be greatly impacted if, for example, a decision was made to deprioritize restoring reefs further than a given proximity to a port or in the far Northern GBR region. The fixed capital costs associated with establishing a facility (e.g., Cairns) deters optimality of subsequent facilities in its proximity (e.g., Townsville); and subsequently, optimality of a facility location rises as distance from an open facility increases. It must be highlighted that this location assessment does not consider other key factors such as the location of existing facilities that may be easily upgraded, or considerations such as the cost and availability of suitable land, utilities, and transport infrastructure. When these factors are considered, the overall optimal location may change.

Different sources of uncertainty have differing impacts on the optimal solution and obtaining certainty of these parameters have associated costs. An assessment of cost and benefit of obtaining parameter certainty needs to be made to assess the robustness of solutions [39]. We showed that certainty of variable (capital and operational) facility costs is not valuable, with large scaling factors ($2^{10}$ and $2^9$, respectively) required to change the optimal solution. The current assumption of homogeneous costs across facility locations may not be realistic. In practice, more heavily populated locations (e.g., Townsville) may be favoured as optimal locations, yielding lower costs with greater access to resources.

Optimal growth time dictates deployment frequency. Production quantity is used to inform total floorspace requirements driving facility sizing. Cost variation impacts the number of facilities, with the correlation based on the parameter varied. Reduction in the optimal number of facilities is primarily driven by raised fixed capital costs, lowered transport costs. Relatively high fixed capital costs deter additional facility sites, but rather expansion of fewer facilities. Reduced transport costs encourage fewer facilities as the weight of transport costs relative to total cost decreases. Deployment costs (a function of distance travelled between facilities and reefs) have a large impact on the number and location of facilities as these operational costs are incurred periodically; whereas facility establishment costs are incurred once but are relatively large.

Deployment costs are simplified as a function of single-mode straight-line distances travelled by a homogeneous fleet of vessels between ports and reef clusters with additional intra-cluster trips. The consideration of dual-mode transport would improve solution accuracy as the relationship of cost to distance becomes more complex. Transport from the facility to its closest port, as well as the consideration of different types of vessels and their various characteristics are not included, nor the use of a barge or large vessel to distribute corals to smaller deployment vessels. Further optimization of reef clustering could reduce distances travelled. These additional transport considerations would improve solution definition and accuracy.

The proportion of coral grown that survive in the wild affects the short- and long-term production decisions, impacting total cost. Reduced time in a facility increases production capacity yet impacts coral survival rates. A higher proportion of coral survival allows production of

fewer corals to meet a given demand for surviving corals. Decreased production causes reduced facility, transport, and operating costs. Assumptions of the relationship between coral survival and growth time in a facility contain significant uncertainty. Greater certainty of coral survival rates with respect to growth time is recommended in order to provide more meaningful, accurate, and reliable results.

Consideration of additional factors would improve the accuracy of the model. Extension of the proposed techniques to consider limited availability and location-specific heterogeneity of transport vessels warrants further investigation in future work. Future facility development and growth considerations optimizing production expansion over time would provide a multi-staged solution to upgrade the existing distribution network based on changing demands [40–42]. Significant variance is observed in the sizing of facilities with respect to variance in coral survival. Therefore, consideration and assessment for building additional facilities, modular expansion of planned facilities, and losses associated with reduced utilization and reduction of existing facilities could be integrated [42]. This consideration would allow strategic planning decisions to be incorporated into plans for large infrastructure and the implications on long-term cost optimality. Further survival considerations of external environmental factors could be incorporated in this model. Seasonal changes throughout the year affect weather conditions including water temperature, storms and cyclones, water currents, and water quality. Proportional survival may be dependent on deployment locations–defined by their proximity to the equator, coastline, continental shelf, and bathymetric features–and environmental conditions at various deployment times, seasons, depths, and locations [26, 33]. Multi-species coral aquaculture production can be modelled, assessed, and analysed for cost optimality with greater certainty of species-specific survival rates.

The proposed model and approach provide a support tool for decision-makers in the production and deployment of grown resources, specifically asexually propagated coral aquaculture. Whilst for the purpose of this study, we have considered land-based aquaculture facilities, our methods could be adapted to investigate increasingly used sea-based in-situ approaches [23]. Future comparisons between different aquaculture strategies may also consider differing objectives, including carbon footprint, costs, and uncertainty of coral survival in less controllable growing conditions. The current model formulation does not consider carbon emissions generated by the establishing and maintaining aquaculture facilities. These costs can be incorporated into the objective function in future analysis to consider these factors.

The value of this contribution is an adaptable decision-making tool. A range of considerations can be incorporated in the future via the objective function and model constraints, such as costs associated with carbon emissions and uncertainty of coral survival under variable growing conditions. This model may be extended to environmental restoration projects across marine and terrestrial applications, forestry, aquaculture of fish and other marine species, and livestock agriculture. With a growing demand and scarce resources available for coral restoration projects, optimizing production is vital to their success. In the era of climate change, this is expected to have growing applications in environmental conservation worldwide as active restoration becomes imperative.

## Supporting information

**S1 Dataset.**
(CSV)

**S2 Dataset.**
(DB)

## Acknowledgments

We acknowledge the Traditional Owners of the sea Country of the Great Barrier Reef where this research is focused, and pay our respects to their Elders past, present, and emerging. We acknowledge their continuing spiritual connection to their sea Country and their ancestors as the first marine scientists.

We acknowledge the Australian Institute for Marine Science for data; Andrea Severati for discussions and advice on coral aquaculture processes and growth rates; Mark Baxendale for Python advice; and Christopher Doropoulos for discussions and advice on coral survival curves and relationships.

## Author Contributions

**Conceptualization:** Ryu B. Lippmann, Kate J. Helmstedt, Paul Corry.

**Data curation:** Ryu B. Lippmann.

**Formal analysis:** Ryu B. Lippmann.

**Funding acquisition:** Mark T. Gibbs, Paul Corry.

**Investigation:** Ryu B. Lippmann.

**Methodology:** Ryu B. Lippmann, Paul Corry.

**Project administration:** Ryu B. Lippmann, Kate J. Helmstedt, Mark T. Gibbs, Paul Corry.

**Resources:** Ryu B. Lippmann, Paul Corry.

**Software:** Ryu B. Lippmann.

**Supervision:** Kate J. Helmstedt, Mark T. Gibbs, Paul Corry.

**Validation:** Ryu B. Lippmann.

**Visualization:** Ryu B. Lippmann.

**Writing – original draft:** Ryu B. Lippmann.

**Writing – review & editing:** Ryu B. Lippmann, Kate J. Helmstedt, Mark T. Gibbs, Paul Corry.

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
