## [Decision Letter · Decision Letter 0]

24 Nov 2022

PONE-D-22-26175Optimizing facility location, sizing, and growth time for a cultivated resource: A case study in coral aquaculturePLOS ONE

Dear Dr. Lippmann,

Thank you for submitting your manuscript to PLOS ONE. After careful consideration, we feel that it has merit but does not fully meet PLOS ONE’s publication criteria as it currently stands. Therefore, we invite you to submit a revised version of the manuscript that addresses the points raised during the review process.

Hello, 

   I have had your article reviewed by two experts in the field, and both have raised some concerns and made some comments that could potentially strengthen the article. I am therefore deeming this a major revision. Were you in agreement with their recommendations, I would welcome the submission of a revised version of the article in the coming weeks. 

Thanks, 

Anderson

We look forward to receiving your revised manuscript.

Kind regards,

Anderson B. Mayfield, Ph.D.

Academic Editor

PLOS ONE

Journal Requirements:

"The work was supported by the Reef Restoration and Adaptation Program, funded by the partnership between the Australian Government’s Reef Trust and the Great Barrier Reef Foundation. 

R.B.L. is supported by the Reef Restoration and Adaptation Program Logistics Scholarship. 

K.J.H. is supported by Australian Research Council Fellowship under Grant DE200101791."

"The work was supported by the Reef Restoration and Adaptation Program, funded by the partnership between the Australian Government’s Reef Trust and the Great Barrier Reef Foundation. 

R.B.L. is supported by the Reef Restoration and Adaptation Program Logistics Scholarship. 

K.J.H. is supported by Australian Research Council Fellowship under Grant DE200101791."

7. We note that Figure 1 in your submission contain map/satellite image which may be copyrighted. All PLOS content is published under the Creative Commons Attribution License (CC BY 4.0), which means that the manuscript, images, and Supporting Information files will be freely available online, and any third party is permitted to access, download, copy, distribute, and use these materials in any way, even commercially, with proper attribution. For these reasons, we cannot publish previously copyrighted maps or satellite images created using proprietary data, such as Google software (Google Maps, Street View, and Earth). For more information, see our copyright guidelines: http://journals.plos.org/plosone/s/licenses-and-copyright.

Additional Editor Comments:

Hello,

I have had your article reviewed by two experts in the field, and both have raised some concerns and made some comments that could potentially strengthen the article. I am therefore deeming this a major revision. Were you in agreement with their recommendations, I would welcome the submission of a revised version of the article in the coming weeks.

Thanks,

Anderson

Reviewers' comments:

Reviewer's Responses to Questions

**Comments to the Author**

1. Is the manuscript technically sound, and do the data support the conclusions?

Reviewer #1: Yes

Reviewer #2: Partly

2. Has the statistical analysis been performed appropriately and rigorously? 

Reviewer #1: Yes

Reviewer #2: I Don't Know

3. Have the authors made all data underlying the findings in their manuscript fully available?

Reviewer #1: Yes

Reviewer #2: Yes

4. Is the manuscript presented in an intelligible fashion and written in standard English?

Reviewer #1: Yes

Reviewer #2: Yes

5. Review Comments to the Author

Reviewer #1: Lippmann and collaborators propose a mathematical programming model that can predict the optimal size and location of coral nurseries based on the survival rate of transplanted corals and subsequent costs. I found the model proposed interesting and well done. Methods were clearly outlined, result and discussion sections covered all necessary points, and publications related to the topic were cited. I believe that the model has significant value for conservation measures, even though age-based survival of transplanted coral is not fully characterized.

The manuscript was enjoyable to read and I only have a few minor edits to make.

Line 57: This sentence needs to be rephrased.

Line 123: Section 0? I don’t understand, is this a typographical error?

Line 397 and Figure 3: Cairns reef cluster needs to be colored differently, since gray is used in the previous figure to indicate reefs excluded from the restoration project.

Line 476: Why is the table included in the supplementary material? It can be incorporated into the main text.

Reviewer #2: PONE-D-22-26175 – Optimizing facility location, sizing, and growth time for a cultivated resource: A case

study in coral aquaculture

This paper approaches large-scale land-based coral aquaculture from an ‘optimal planning’ perspective as applied generally to cultivated – as opposed to manufactured – resources. The concept is interesting and worthy of exploration. I reviewed the manuscript from the angle of my training as a biologist and not an expert on supply chain logistics, computational mathematics, or engineering. In full disclosure, most of the Methods and Results described here are not familiar to me. Thus, other reviewers should be relied upon to comment on those technical and theoretical aspects of the paper. While I believe the approach is novel and clever and the manuscript is well-written, I have some concerns with the general concepts expressed in the paper. These are detailed below.

General comments:

- While I am a proponent of culturing corals for reef restoration, I (and I believe many others in this community) remain unconvinced and skeptical that large-scale land-based culture is ultimately a viable way to produce coral biomass for outplanting. The utility is apparent for controlled sexual reproduction or “genetic banking” of living corals. However, the scalability issues that plague restoration (e.g. this manuscript is considering outplanting at 50 out of 2816 reef in the GBR) are magnified in land-based systems. While I have no comparative data to support this assertion, I have experience working and conducting research in both land-based and ocean-based coral nurseries, and my intuition is that – unless a land-based nursery can be run almost entirely on renewable energy – the carbon footprint of producing coral biomass in a land-based system is prohibitive. Land-based coral culture is also extremely technically difficult, and corals tend to grow faster in ocean-based nurseries. Given this admitted bias, I do believe the authors have done valuable work here. To support publication, I would need to see acknowledgement in the manuscript that land-based facilities are not the only means of producing coral biomass for restoration, some “disclaimer” in terms of the amount of excess carbon dioxide emissions that may be created by industrial-scale coral production on land, and some discussion of these considerations.

- While I see the utility of the tool and the authors do certainly acknowledge that many assumptions are included in the calculations, in my opinion this could be highlighted to an even greater extent. The specific case study model could be updated as more data become available to fill in the values. Conclusions such as Cairns making the most sense for a facility due to location appear valid due to low sensitivity to costing/growth/survival variables. However for now, it is my opinion that the specific dollar values, required square meters, etc that are model outputs should be taken only as estimates in the broadest sense. Coral growth and survival both in aquaculture and after outplanting can be highly variable and system/location dependent. Overall, the authors do a good job of discussing things in relative terms and directional effects and magnitude of impacts from the different input variables, so only suggesting some additional acknowledgement of the level of uncertainty inherent in specific case study outputs.

Specific comments:

lines 40 - 44 – While on a broad scale this assumption (the longer/larger you grow something the better it will survive upon release) makes sense, in my opinion there should at least be some acknowledgement that this is a necessary oversimplification. An example from reforestation is cited but transitioning to corals complicates things a bit. Unlike trees, corals are often fragmented before being deployed to the reef, so even the basic assumption that the longer you grow something the larger it will be does not always hold up. Further, the below paper offers an example of a scenario in which corals grown longer on land to a larger size were outperformed (in terms of both post-outplant growth and survival) by the same species grown on land for a shorter period of time to a smaller size:

Henry, J.A., O’Neil, K.L., Pilnick, A.R. and Patterson, J.T., 2021. Strategies for integrating sexually propagated corals into Caribbean reef restoration: experimental results and considerations. Coral Reefs, 40(5), pp.1667-1677.

Line 57 – Suggest deleting “Cultivating”.

Lines 60-62 – Why just on land? Currently, most coral biomass for use in reef restoration is produced in ocean-based nurseries. Much more published data exists on coral growth rates in ocean-based than in land-based facilities.

Lines 88-91 – Could carbon emissions and other environmental impacts be included in this analysis? Greenhouses and enclosed structures used to grow corals require tremendous amounts of energy to pump, heat, and cool water and air as well as other ancillary processes. An analysis of the carbon footprint per unit of coral biomass produced in a land-based system would be very interesting and informative to the question of whether it makes sense to be producing coral biomass for outplanting in land-based systems.

6. PLOS authors have the option to publish the peer review history of their article (what does this mean?). If published, this will include your full peer review and any attached files.

Reviewer #1: No

Reviewer #2: No

---

## [Author Response · Author response to Decision Letter 0]

10 Jan 2023

We thank the two Reviewers for their thorough and insightful reviews. We have largely implemented the changes, improving the manuscript. Responses to each comment follow. In this review, we also made some minor grammatical and typographical changes that can be seen in the manuscript containing tracked changes.

Revisions in the manuscript are shown in red text.

Reviewer #1: 

General comments: Lippmann and collaborators propose a mathematical programming model that can predict the optimal size and location of coral nurseries based on the survival rate of transplanted corals and subsequent costs. I found the model proposed interesting and well done. Methods were clearly outlined, result and discussion sections covered all necessary points, and publications related to the topic were cited. I believe that the model has significant value for conservation measures, even though age-based survival of transplanted coral is not fully characterized.

The manuscript was enjoyable to read and I only have a few minor edits to make.

We thank the Reviewer for their valuable comments and improvements. Revisions have been made to address their comments.

Comment 1

Line 57: This sentence needs to be rephrased.

This sentence at lines 51-53 has been rephrased.

[…] Planning how to best use resources for aquaculture projects to cultivate coral for large-scale reef restoration is vital to efficiently and sustainably deploying grown corals [21] […]

Comment 2

Line 123: Section 0? I don’t understand, is this a typographical error?

Yes, this was a typographical/formatting error, thanks for catching it. Now, at lines 136-137 reads:

[…] section 2.2 […]

Comment 3

Line 397 and Figure 3: Cairns reef cluster needs to be colored differently, since gray is used in the previous figure to indicate reefs excluded from the restoration project.

We have changed the coloring of reefs attributed to Cairns from gray to purple, and the legend for Fig 3 has also been appropriately updated.

Fig 3. A spatial plot of optimal solution for parametrized base case. Reef clusters are sized by demand quantity and coloured by optimal port assignment – Cairns in purple, Townsville in brown, Airlie Beach in orange, Mackay in green, and Rockhampton in red. Optimal facility locations are coloured black sized by their production quantity, and indicated as proportion of total coral produced.

Comment 4

Line 476: Why is the table included in the supplementary material? It can be incorporated into the main text.

We agree with the Reviewer’s suggestion and have amended the main text accordingly. This has been renamed Table 3 and moved from the Supplementary Information into Section 3.2.3. 

Reviewer #2: 

This paper approaches large-scale land-based coral aquaculture from an ‘optimal planning’ perspective as applied generally to cultivated – as opposed to manufactured – resources. The concept is interesting and worthy of exploration. I reviewed the manuscript from the angle of my training as a biologist and not an expert on supply chain logistics, computational mathematics, or engineering. In full disclosure, most of the Methods and Results described here are not familiar to me. Thus, other reviewers should be relied upon to comment on those technical and theoretical aspects of the paper. While I believe the approach is novel and clever and the manuscript is well-written, I have some concerns with the general concepts expressed in the paper. These are detailed below.

General comments:

- While I am a proponent of culturing corals for reef restoration, I (and I believe many others in this community) remain unconvinced and skeptical that large-scale land-based culture is ultimately a viable way to produce coral biomass for outplanting. The utility is apparent for controlled sexual reproduction or “genetic banking” of living corals. However, the scalability issues that plague restoration (e.g. this manuscript is considering outplanting at 50 out of 2816 reef in the GBR) are magnified in land-based systems. While I have no comparative data to support this assertion, I have experience working and conducting research in both land-based and ocean-based coral nurseries, and my intuition is that – unless a land-based nursery can be run almost entirely on renewable energy – the carbon footprint of producing coral biomass in a land-based system is prohibitive. Land-based coral culture is also extremely technically difficult, and corals tend to grow faster in ocean-based nurseries. Given this admitted bias, I do believe the authors have done valuable work here. To support publication, I would need to see acknowledgement in the manuscript that land-based facilities are not the only means of producing coral biomass for restoration, some “disclaimer” in terms of the amount of excess carbon dioxide emissions that may be created by industrial-scale coral production on land, and some discussion of these considerations.

We thank the Reviewer for these important points. We agree that these were overlooked in our previous submission. We have amended the manuscript to ensure that these issues outlined by the Reviewer, as well as other relevant limitations and considerations, are identified and discussed in multiple places throughout the manuscript. We now clarify that this study is focused on the optimization of land-based aquaculture methods.

Acknowledgement of the aquaculture approaches to produce biomass has been included.

• In the introduction at lines 56-73:

There are various approaches to using aquaculture to grow young corals to produce coral biomass for restoration. The nature of this planning problem is dependent on the specific style of aquaculture carried out. Corals can be propagated sexually through larval spawning or asexually by fragmentation. Whilst sexually propagated corals offer the potential to enhance genetic diversity of reefs, further considerations must be made to consider the timing and logistics of coral spawning [8]. Asexual propagation involves pre-settling fragmented corals onto deployment devices, and rearing corals until deployed. This allows greater control over production planning and scheduling. The placement of aquaculture facilities on land or in the sea is another important factor considered in these projects. In-situ approaches grow coral fragments or larvae in sea-based nurseries [23]. Land-based ex-situ aquaculture involves cultivating corals in a controlled environment on land before transporting and deploying young corals to host reef sites [22,24]. This approach can lead to high quality control and throughput but requires establishment and operation of land-based facilities. Land-based aquaculture generates significant energy demands for pumping water and sustaining essential utilities. These energy demands, unless met by renewable sources, produce carbon emissions, contributing to climate change and reef degradation, reinforcing the degradation that the intervention is aiming to combat [25]. Operations need to be carefully assessed, optimized, and managed to address these issues.

• In the Methods at lines 111-116, this distinction in our consideration of only one specific restoration method is made further explicit:

We propose a model that considers the construction and operation of land-based ex-situ coral aquaculture facilities over a prescribed planning horizon to cultivate and deploy corals to specified reefs in the wild. We then use this model to optimize the size and placement of the facilities to minimize amortized costs and meet demand for reef adaptation and restoration outcomes. Here we assume that planners have specified the use of land-based facilities rather than comparing different aquaculture strategies. […]

• In the Discussion at lines 623-627, the caveat is reinforced:

The proposed model and approach provide a support tool for decision-makers in the production and deployment of grown resources, specifically asexually propagated coral aquaculture. Whilst for the purpose of this study, we have considered land-based aquaculture facilities, our methods could be adapted to investigate increasingly used sea-based in-situ approaches [23]. […]

We address carbon emissions and energy consumption in response to Comment 4.

- While I see the utility of the tool and the authors do certainly acknowledge that many assumptions are included in the calculations, in my opinion this could be highlighted to an even greater extent. The specific case study model could be updated as more data become available to fill in the values. Conclusions such as Cairns making the most sense for a facility due to location appear valid due to low sensitivity to costing/growth/survival variables. However for now, it is my opinion that the specific dollar values, required square meters, etc that are model outputs should be taken only as estimates in the broadest sense. Coral growth and survival both in aquaculture and after outplanting can be highly variable and system/location dependent. Overall, the authors do a good job of discussing things in relative terms and directional effects and magnitude of impacts from the different input variables, so only suggesting some additional acknowledgement of the level of uncertainty inherent in specific case study outputs.

We have reinforced our discussion by adding a disclaimer about uncertainty.

• In the Methods at lines 178-181:

[…] The cost parametrization is outlined in Table 1. These cost estimates are based on expert elicitation and estimates for facilities that have not yet been built. As a result, these estimates have high associated uncertainty and should be refined for a particular circumstance and time if plans are being developed. 

Specific comments:

Comment 1

lines 40 - 44 – While on a broad scale this assumption (the longer/larger you grow something the better it will survive upon release) makes sense, in my opinion there should at least be some acknowledgement that this is a necessary oversimplification. An example from reforestation is cited but transitioning to corals complicates things a bit. Unlike trees, corals are often fragmented before being deployed to the reef, so even the basic assumption that the longer you grow something the larger it will be does not always hold up. Further, the below paper offers an example of a scenario in which corals grown longer on land to a larger size were outperformed (in terms of both post-outplant growth and survival) by the same species grown on land for a shorter period of time to a smaller size:

Henry, J.A., O’Neil, K.L., Pilnick, A.R. and Patterson, J.T., 2021. Strategies for integrating sexually propagated corals into Caribbean reef restoration: experimental results and considerations. Coral Reefs, 40(5), pp.1667-1677.

As the Reviewer points out here, we have been unclear about the assumed relationship of coral growth time with the chance of survive upon release. We agree that coral science has not fully solved this problem and we do not have one proven relationship between time and survival. This uncertainty is what motivated our exploration of various survival curves – one of which decreases beyond a maximum proportional survival rate. We have added additional clarification and explanation to clarify this assumption and uncertainty. 

• In the introduction at lines 28-30:

[…] Whilst this oversimplification of growth time being correlated with proportional survival is acknowledged, we assume the existence of a relationship [8]. […]

where [8] is a citation of the paper suggested by the Reviewer.

This has brought a miscommunication to our attention – we were not clear enough that the reproduction we model here always begins with fragmentation. The growth times are times after fragmentation has occurred. We have clarified this by adding:

• In the introduction at lines 74-75:

In designing such facilities, the growth time allocated for corals after being fragmented and settled in facilities is considered. […]

Although we only consider time as a factor for coral survival, it is a good point that this growth and survival would also be dependent on other factors e.g., deployment site/location temperature, topography, light, depth. Whilst these are not considered in this study, we have amended the manuscript to further acknowledge their potential implications 

• In the Discussion at lines 614-620:

[…] Further survival considerations of external environmental factors could be incorporated in this model. Seasonal changes throughout the year affect weather conditions including water temperature, storms and cyclones, water currents, and water quality. Proportional survival may be dependent on deployment locations – defined by their proximity to the equator, coastline, continental shelf, and bathymetric features – and environmental conditions at various deployment times, seasons, depths, and locations [26,33]. […]

Comment 2

Line 57 – Suggest deleting “Cultivating”.

Yes, agreed. We have deleted this and amended the manuscript following the Reviewer’s suggestion.

Comment 3

Lines 60-62 – Why just on land? Currently, most coral biomass for use in reef restoration is produced in ocean-based nurseries. Much more published data exists on coral growth rates in ocean-based than in land-based facilities.

We have included a summary of aquaculture options and coral propagation methods.

• In the Introduction at lines 58-69:

[…] Corals can be propagated sexually through larval spawning or asexually by fragmentation. Whilst sexually propagated corals offer the potential to enhance genetic diversity of reefs, further considerations must be made to consider the timing and logistics of coral spawning [8]. Asexual propagation involves pre-settling fragmented corals onto deployment devices, and rearing corals until deployed. This allows greater control over production planning and scheduling. The placement of aquaculture facilities on land or in the sea is another important factor considered in these projects. In-situ approaches grow coral fragments or larvae in sea-based nurseries [23]. Land-based ex-situ aquaculture involves cultivating corals in a controlled environment on land before transporting and deploying young corals to host reef sites [22,24]. This approach can lead to high quality control and throughput but requires establishment and operation of land-based facilities. […]

We have added a brief, but pertinent disclaimer to clarify the purpose of this research.

• In the Introduction at lines 105-107

[…] This study does not advocate for a specific aquaculture method, but rather optimizes the location and design of land-based aquaculture facilities in applications where they are being considered. […]

Comment 4

Lines 88-91 – Could carbon emissions and other environmental impacts be included in this analysis? Greenhouses and enclosed structures used to grow corals require tremendous amounts of energy to pump, heat, and cool water and air as well as other ancillary processes. An analysis of the carbon footprint per unit of coral biomass produced in a land-based system would be very interesting and informative to the question of whether it makes sense to be producing coral biomass for outplanting in land-based systems.

Whilst the carbon footprint would certainly be smaller for in-situ aquaculture, here we explicitly consider ex-situ aquaculture. We do not intend to imply that this method is better by having a smaller carbon footprint, lower cost, or other measure; this method is simply the focus of this assessment. Since the coral aquaculture field is rapidly developing without much current or past implementation, there are not currently good estimates of the carbon footprint in the literature. This speaks to the Reviewer’s earlier comment about uncertainty around costs too – the estimates of carbon footprint numbers are likely to be even larger, and out of scope for this decision problem. As stated in response to the Reviewer’s general comments, we have amended the manuscript to include these considerations.

We have included a disclaimer regarding the assessment of implications and limitations of the model, including carbon emissions highlighted by the Reviewer, and emphasised the high energy demands associated with land-based aquaculture.

• In the introduction at lines 69-73:

[…] Land-based aquaculture generates significant energy demands for pumping water and sustaining essential utilities. These energy demands, unless met by renewable sources, produce carbon emissions, contributing to climate change and reef degradation, reinforcing the degradation that the intervention is aiming to combat [25]. Operations need to be carefully assessed, optimized, and managed to address these issues.

• In the Discussion at lines 627-636:

[…] Future comparisons between different aquaculture strategies may also consider differing objectives, including carbon footprint, costs, and uncertainty of coral survival in less controllable growing conditions. The current model formulation does not consider carbon emissions generated by the establishing and maintaining aquaculture facilities. These costs can be incorporated into the objective function in future analysis to consider these factors.

The value of this contribution is an adaptable decision-making tool. A range of considerations can be incorporated via the objective function and model constraints, such as costs associated with carbon emissions and uncertainty of coral survival under variable growing conditions. […]

As the field and published estimates do develop, we would hope to perform an analysis including multiple objectives including environmental.

• In the Discussion at lines 625-632: 

[…] Whilst for the purpose of this study, we have considered land-based aquaculture facilities, our methods could be adapted to investigate increasingly used sea-based in-situ approaches [23]. Future comparisons between different aquaculture strategies may also consider differing objectives, including carbon footprint, costs, and uncertainty of coral survival in less controllable growing conditions. The current model formulation does not consider carbon emissions generated by the establishing and maintaining aquaculture facilities. These costs can be incorporated into the objective function in future analysis to consider these factors.

---

## [Decision Letter · Decision Letter 1]

20 Feb 2023

Optimizing facility location, sizing, and growth time for a cultivated resource: A case study in coral aquaculture

PONE-D-22-26175R1

Dear Dr. Lippmann,

We’re pleased to inform you that your manuscript has been judged scientifically suitable for publication and will be formally accepted for publication once it meets all outstanding technical requirements.

Kind regards,

Anderson B. Mayfield, Ph.D.

Academic Editor

PLOS ONE

Additional Editor Comments (optional):

Hello,

Both reviewers found your responses to their concerns satisfactory. As such, I am pleased to now accept this article.

Anderson

Reviewers' comments:

Reviewer's Responses to Questions

**Comments to the Author**

1. If the authors have adequately addressed your comments raised in a previous round of review and you feel that this manuscript is now acceptable for publication, you may indicate that here to bypass the “Comments to the Author” section, enter your conflict of interest statement in the “Confidential to Editor” section, and submit your "Accept" recommendation.

Reviewer #1: All comments have been addressed

Reviewer #2: All comments have been addressed

2. Is the manuscript technically sound, and do the data support the conclusions?

Reviewer #1: (No Response)

Reviewer #2: Yes

3. Has the statistical analysis been performed appropriately and rigorously? 

Reviewer #1: (No Response)

Reviewer #2: Yes

4. Have the authors made all data underlying the findings in their manuscript fully available?

Reviewer #1: (No Response)

Reviewer #2: Yes

5. Is the manuscript presented in an intelligible fashion and written in standard English?

Reviewer #1: (No Response)

Reviewer #2: Yes

6. Review Comments to the Author

Reviewer #1: (No Response)

Reviewer #2: (No Response)

7. PLOS authors have the option to publish the peer review history of their article (what does this mean?). If published, this will include your full peer review and any attached files.

Reviewer #1: No

Reviewer #2: No

---

## [Editor Report · Acceptance letter]

6 Mar 2023

PONE-D-22-26175R1 

Optimizing facility location, sizing, and growth time for a cultivated resource: A case study in coral aquaculture 

Dear Dr. Lippmann:

I'm pleased to inform you that your manuscript has been deemed suitable for publication in PLOS ONE. Congratulations! Your manuscript is now with our production department. 

Kind regards, 

on behalf of

Dr. Anderson B. Mayfield 

Academic Editor

PLOS ONE